# An Improved Asphalt Penetration Test Method

**DOI:** 10.3390/ma14010147

**Published:** 2020-12-31

**Authors:** Zhoujing Ye, Wenjuan Ren, Hailu Yang, Yinghao Miao, Fengyan Sun, Linbing Wang

**Affiliations:** 1National Center for Materials Service Safety, University of Science and Technology Beijing, Beijing 100083, China; yezhoujing@ustb.edu.cn (Z.Y.); yanghailu@ustb.edu.cn (H.Y.); miaoyinghao@ustb.edu.cn (Y.M.); fysun@ustb.edu.cn (F.S.); 2Shenzhen Oceanpower New Material Technology Co. LTD, Shenzhen 518042, China; wenjuan_ren@163.com; 3Joint USTB-Virginia Tech Lab on Multifunctional Materials, USTB, Virginia Tech, Blacksburg, VA 24061, USA

**Keywords:** asphalt penetration test, high-speed camera, image processing, finite element method

## Abstract

A traditional penetration test only measures the total penetration within 5 s. The penetration process is not monitored, and therefore, a large amount of information on the deformation properties of asphalt is not used. This paper documents a study to use a high-speed camera to quantify the entire penetration process and use the Finite Element Method (FEM) to interpret the penetration process using a viscoelastic model. The penetration–time relationships of several asphalt binders (70#, 90#, a rubber modified binder, and a styrene-butadiene-styrene (SBS) modified binder) have been acquired using the new method, and the FEM modeling of the penetration processes is performed. The results show that both stress relaxation and creep appear during the penetration process. The results indicate that the improved test method and its data interpretation procedure may better characterize the properties of asphalt binder, which may extend the applications of the traditional penetration test.

## 1. Introduction

The properties of asphalt materials affect the service life of asphalt pavement. Under different temperatures, load, and time, asphalt shows different properties, including viscous deformation, elastic deformation, and viscoelasticity [1]. The evaluation index of asphalt performance consists of penetration, softening point, and ductility. Among them, penetration indicates the softness and consistency of asphalt and its ability to resist shear failure. The penetration test has been widely used to test the temperature sensitivity [2,3,4,5,6], rheology [7,8,9,10,11,12], and diffusion properties of asphalt materials [13].

However, a traditional penetration test only measures the total penetration within 5 s. The penetration process is not monitored. A large amount of information on the deformation properties of asphalt is not used. Additionally, the asphalt penetration test is affected by many factors. Changes in the test condition and operation will result in a significant variation in the penetration index (PI) [14,15,16,17,18]. The influencing factors include the material and weight of the standard needle; the cooling, holding, and heating time of the sample; the material and size of the sample container; the release agent used and the operation of tester.

In order to solve the above-mentioned problems, the repeated test, comparison test, and simulation test were used to reduce the error of the penetration test and improve the efficiency of the asphalt penetration test. Zhang et al. [19] studied the influence of the difference among laboratory equipment, environment, and personnel on the asphalt penetration test results, thereby enhancing the test detection level and the accuracy of the data. Sun et al. [20] conducted a penetration test on SBS modified asphalt with different dosages and types of SBS modifier under three penetration weights and four different temperatures and then proposed a reasonable penetration test method for evaluating the performance of SBS modified asphalt, which can better evaluate the temperature sensitivity of this kind of asphalt. Sun et al. [21] developed an asphalt virtual simulation experiment based on flash technology, including the asphalt penetration test. The demonstration mode provides the standard operation video, while the operation mode realizes the whole operation process of the real experiment and evaluates the accuracy of the virtual experiment operation results. The above method improves the asphalt penetration test and reduces the test error but lacks the analysis on the whole process.

In order to monitor the penetration process and analyze the information of the deformation properties of asphalt, the high-speed photography technology and the finite element simulation method are adopted to record and simulate the penetration process of the asphalt. Then, the creep and stress relaxation phenomena that occur in the asphalt during this process can be intuitively analyzed, which is of great significance for evaluating the properties of asphalt.

## 2. Asphalt Penetration Test

### 2.1. Test Design

The SYD-2801E1 electronic automatic penetrometer produced by Shanghai Changji Geological Instrument Co., Ltd was used in the test, and the procedures of the standard penetration test are shown below:(1)Heat four different types of asphalt (70# matrix asphalt, 90# matrix asphalt, SBS modified asphalt, and rubber modified asphalt) selected for the test.(2)Put the heated asphalt into the 35 × 55 mm sample container and make asphalt samples.(3)Cool the prepared asphalt samples at room temperature for 1.5 h.(4)Put the samples into the thermostat water bath preset at 25 °C (the normal temperature condition) for 1.5 h, as shown in Figure 1.(5)Take the sample out of the water bath, put it into the glassware of asphalt penetration instrument, keep the water temperature at 25 °C, and then perform the asphalt penetration test.

In this experiment, the relationship curve between the penetration value and time was obtained with the high-speed camera shooting method. The model of the high-speed camera was Olympus i-SPEED 3 (Olympus Corporation, Tokyo, Japan). The high-speed shooting can reach 150,000 fps, with the advantages of high resolution and low light sensitivity. The specific test parameters are shown in Table 1.

In order to record the vertical displacement of the needle during the test, the needle connecting rod part was used as the shooting target object and kept in the same straight line with the lens of the high-speed camera. In addition, the reference was a straightedge fixed on the automatic penetrometer. The test process is shown in Figure 2.

70# matrix asphalt, 90# matrix asphalt, SBS modified asphalt, and rubber modified asphalt were tested for penetration, respectively. Three parallel tests were carried out on the same asphalt, and the standard needle was replaced, once for each test. The final result was the average of three test values to minimize errors caused by operating conditions.

### 2.2. Image Acquisition and Processing

Through i-SPEED Viewer software (version 22.0.3.2), the videos were converted into 1500 images in the JPG format, and then a total of 6 images were taken out at a ratio of 1/300 to directly observe the vertical displacement of the standard needle, as shown in Figure 3.

Through the MATLAB program (R2017a), the color picture was converted to grayscale, thus improving the efficiency of pixel extraction. The needle-connecting rod part was positioned as the feature point. The straightedge was used as a reference scale to calculate the vertical displacement of the standard needle. Through image processing, the length of every 35 pixels was equal to the actual distance of 1 cm. The gray value of 5 pixels at the needle connecting rod part was extracted as the feature point. The image processing is displayed in Figure 4.

A set of data was selected every 30 frames in this test for needle penetration displacement analysis so as to reflect the change of displacement more accurately.

As shown in Figure 5, in the range of 0–300 fps, the asphalt penetration value shows an overall upward trend because the standard needle moves in free fall when it first descends, and the displacement changes obviously. In the range of 300–600 fps, the penetration values of rubber modified asphalt and SBS modified asphalt gradually increase with time, and the maximum increase range is about 0.5 mm. The penetration value of 70# matrix asphalt experiences a slight oscillation, with the fluctuation range within 1.2 mm. The penetration of 90# matrix asphalt is featured with a relatively dramatic oscillation, with the maximum fluctuation amplitude up to 1.8 mm. In the range of 700–1200 fps, 90# matrix asphalt still has a small vibration and presents an overall upward trend. The other three kinds of asphalt show a slow upward trend. In the range of 1200–1500 fps, the penetration values of the four types of asphalt gradually stabilize and tend to be constant.

From the above phenomenon, 90# matrix asphalt has weak recovery ability. When the standard needle penetrates 90# matrix asphalt, it first produces certain displacement. Due to the influence of shear force, the internal structure of the asphalt changes, which leads to obvious fluctuations. According to the fluctuation range and trend of the penetration value, the stability of SBS modified asphalt and rubber modified asphalt is better than that of 70# matrix asphalt and 90# matrix asphalt.

### 2.3. Dynamic Shear Rheological Test

In order to further verify the results of the penetration test, the four kinds of asphalts were subjected to dynamic shear rheological tests with an Anton Paar MCR302 multi-functional rheometer (Anton Paar GmbH, Graz, Austria), respectively. Two parallel plates with a diameter of 25 mm were used, and the distance between plates was 1 mm. Four kinds of asphalts were tested by frequency scanning at 25 °C, and the angular frequency was 0.1~100 rad/s. Among them, the strain control loading was adopted with a value of 1%. In this test, two sets of parallel tests were carried out for each asphalt sample, and the average value was used. Figure 6 shows the variation of rheological parameters G’, G*/sinσ, and tanσ at the same temperature but different frequencies.

In Figure 6, G’ and G*/sinσ of the four kinds of asphalts show an upward trend with the increase of angular frequency. With the increase of the angular frequency, the phase angle of the four asphalts decreases gradually. In terms of asphalt viscoelasticity, rubber modified asphalt has the best elastic property and strong recovery ability. In contrast, 90# matrix asphalt has high viscosity and strong high temperature rheological property, but poor recovery ability. In the process of the dynamic shear rheological test, the viscosity and elasticity of asphalt material were quantified, which was consistent with the analysis results of the penetration test. This explained the variation of the penetration value of asphalt taken by the high-speed camera.

## 3. Asphalt Penetration Simulation

The geometric modeling of the asphalt penetration test was divided into two parts, which were the standard needle model and the asphalt model.

### 3.1. Standard Needle Model

The size of standard needle model is provided in the Test Regulation for Asphalt and Asphalt Mixture for Highway Engineering (JTG E20-2011). The tip of the standard needle was assumed to a hemispherical shape to avoid the deviation of simulation results caused by stress concentration. Since the deformation of the standard needle was much smaller than that of the asphalt, the standard needle was set as a rigid body. The specific parameters are shown in Table 2.

### 3.2. Asphalt Sample Model

70# matrix asphalt sample was selected for finite element simulation; the specific parameters are shown in Table 3.

The size of the asphalt sample model is consistent with the test size, which was φ55 mm and had a depth of 35 mm. The Young’s modulus of asphalt was set as 1700 KPa, Poisson’s ratio was 0.35, and the viscoelasticity of asphalt was simulated by Prony series. The bottom of the asphalt sample was restrained in three directions. The standard needle and symmetrical boundary were restrained in the horizontal direction. The element type was the four-node linear reduction integral unit CAX4R. Figure 7 shows the 360° scan of the asphalt penetration model.

In this case, the standard needle performed a free fall motion during the simulation process; the fall time was set as 5 s and the gravity was set as 9.8 N/kg. The contact between standard needle and asphalt sample was set as surface-to-surface contact, and the friction coefficient was set to zero.

### 3.3. Model Validation

The simulation results of 70# matrix asphalt were compared with the measured results. The point a0 in Figure 8 was selected to obtain the relationship between the penetration value and the time. Figure 9 shows the comparison results.

Although the Finite Element Method (FEM) model was a simplification of the actual situation, the displacement obtained by the two tests had the same trends with the time; the simulation method could reflect the change of the asphalt penetration value in a certain extent. The reason for the difference is that there was a deviation between the fitted viscoelastic parameters and the real value. Second, the temperature and personnel operation in the sample process may cause instability of the results.

## 4. Result Analysis

Figure 8 shows the selected monitoring points, namely a0–a4, and b0-b4. The displacement, stress, and strain of each monitoring point were analyzed during the asphalt penetration test.

### 4.1. Analysis on Surface Monitoring Points

a0 (b0) is the contact point between the needle tip and the asphalt surface. Figure 10 shows the time variation curves of displacement, stress, and strain of a0.

In Figure 10, the needle displacement increases significantly during 0–1 s. At 0.9s, the vertical displacement increases from 0 mm to 5.764 mm. Then, the displacement appears with the vibration phenomenon at 1–3 s. The displacement is 5.696 mm at 2.9 s, which shows unobvious change compared with that at 0.9 s. After 3 s, the displacement increases slowly and reaches 6.799 mm at 3.5 s. At 4–5 s, the displacement reaches the maximum value of 7.304 mm and tends to be constant.

According to the stress–strain curve with time, the stress–strain increases and reaches the maximum value at 0–0.4 s. At 0.4–3 s, for the stress, a certain amplitude of oscillation and instability appear, presenting a downward trend. Meanwhile, the strain of asphalt fluctuates, which reflects the rheological properties of asphalt. At 3–4 s, the stress oscillation weakens, and the strain increases with time. At 4–5 s, the stress gradually decreases, while the strain tends to be stable.

### 4.2. Analysis on Vertical Monitoring Points

Figure 11 shows the time variation curves of displacement, stress, and strain of the vertical monitoring points a1–a4.

In Figure 11, the displacement, stress, and strain of the monitoring point decrease as the depth increases. At 0–1.2 s, the displacement, stress, and strain of each monitoring point change significantly. The stress and strain reach the maximum at 0.3 s, while the increase amplitude of displacement is not obvious. At 0.3–1.2 s, the stress and strain obviously fluctuate up and down, and the displacement rapidly increases to the maximum value. The elastic deformation occurs in this stage, and the maximum stress of elastic deformation is maintained. At 1.2–3 s, the displacement of each monitoring point fluctuates up and down, where a1–a3 are more obvious. At the same time, the stress and strain remain stable without obvious fluctuation. As the loading time increases, the structure of asphalt changes under the action of force, which leads to the instability inside asphalt and the phenomenon that the displacement first decreases and then increases. After 3 s, the displacement of each monitoring point gradually stabilizes and tends to be a constant value. Moreover, the stress gradually decreases, and the strain is featured with an increase trend.

### 4.3. Analysis on Transverse Monitoring Points

Figure 12 shows the time variation curves of displacement, stress, and strain of the transverse monitoring points b1–b4.

In Figure 12, there is no instantaneous and large displacement at each transverse monitoring point during 0–0.3 s, but the stress and strain increase significantly and reach the maximum value. At 0.3–0.7 s, the displacement increases dramatically and reaches the maximum, while the stress and strain remain stable. Compared with the vertical monitoring point, the displacement of the transverse monitoring point reaches the maximum first. In this case, when the standard needle penetrates the asphalt sample, it first produces relatively large displacement in the transverse direction, and then the displacement increases significantly in the vertical direction. In addition, the stress and strain of the transverse and vertical monitoring points basically reach the maximum at the same time. With the increase of time, in all the displacement, stress, and strain of each monitoring point in the transverse direction, the oscillatory instability phenomenon appears. Compared with the vertical direction, the change trends of displacement, stress, and strain concerning the transverse monitoring points are relatively gentle. After 4 s, the displacement and stress changes tend to be stable, while the strain of some monitoring points is inclined to increase.

To sum up, the asphalt penetration test can be considered as a continuous loading process. In the initial stage, the surface of asphalt was featured with a large deformation, and the stress–strain inside the asphalt increased with time. As the loading time increased, the internal structure of the asphalt changed under the action of the force, which caused the fluctuation of displacement, stress, and strain. Since the load continued to be applied to the asphalt, the displacement tended to be a constant value. The stress tended to decrease, while the strain tended to increase, which showed the phenomena of stress relaxation and creep.

## 5. Conclusions

1. The improved penetration test using high-speed imaging and FEM modeling has enabled better data interpretation of the traditional penetration test. It may revive the use of the penetration test as a regular test for asphalt binder characterization.

2. The weaker vibrations during the penetration into rubber modified asphalt and SBS modified asphalt may indicate a strong material structure of the two binders in comparison with the two base asphalt binder 70# and 90#.

3. The viscosity and elasticity of the four kinds of asphalts were quantified in the process of the dynamic shear rheological test, which was consistent with the analysis results of the penetration test. In other words, the modified penetration test may have the capability to characterize viscosity and modulus of asphalt binder.

4. It is anticipated that subsequent research to expand the test temperature range, penetration time and applied load, and to improve the FEM model of asphalt penetration may allow the test to become a comprehensive test for asphalt binder.

## Figures and Tables

**Figure 1 materials-14-00147-f001:**
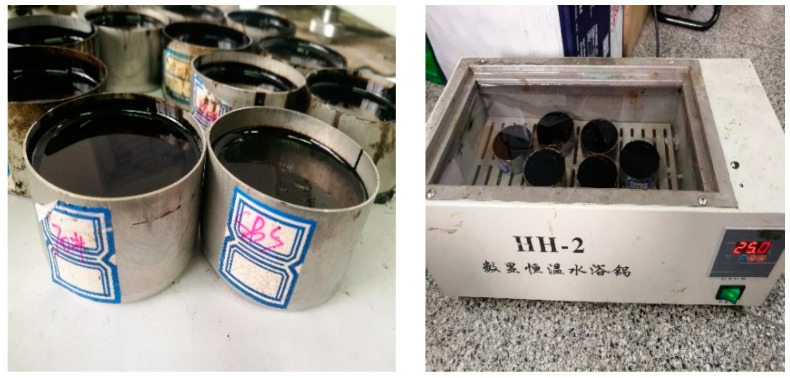
Asphalt samples and the process of the water bath.

**Figure 2 materials-14-00147-f002:**
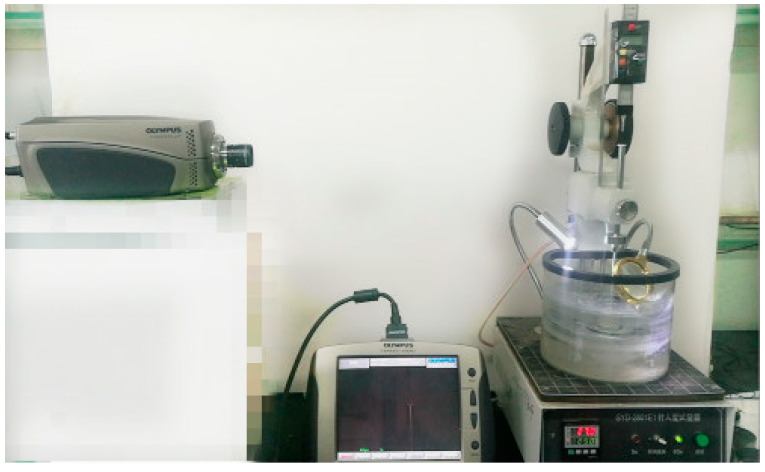
The penetration test process recorded by the high-speed camera.

**Figure 3 materials-14-00147-f003:**
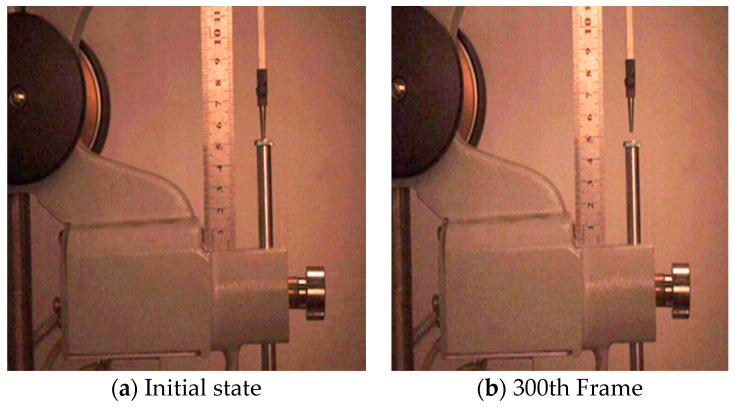
Converting video to JPG images: (**a**) Initial state; (**b**) 300th frame; (**c**) 600th frame; (**d**) 900th frame; (**e**) 1200th frame; (**f**) 1500th frame.

**Figure 4 materials-14-00147-f004:**
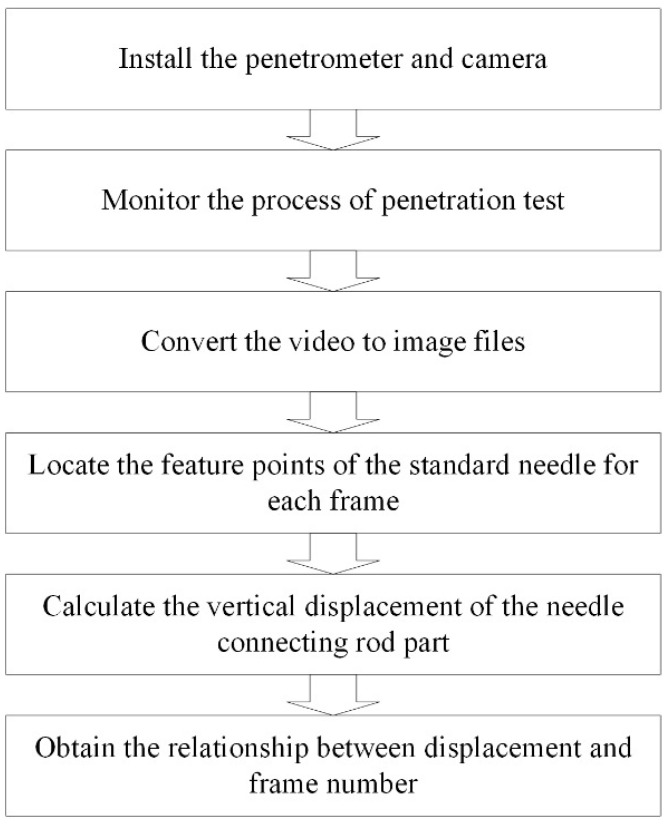
The image processing.

**Figure 5 materials-14-00147-f005:**
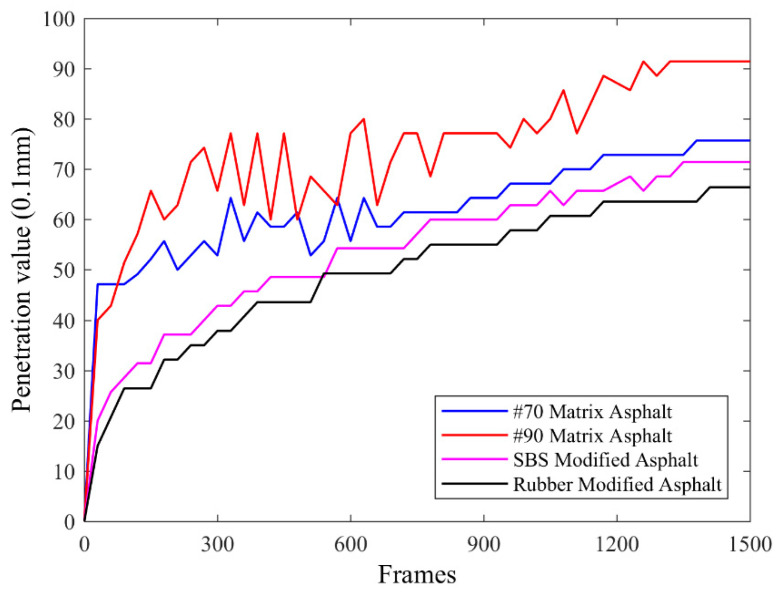
The relationship between the penetration value and the image frame number.

**Figure 6 materials-14-00147-f006:**
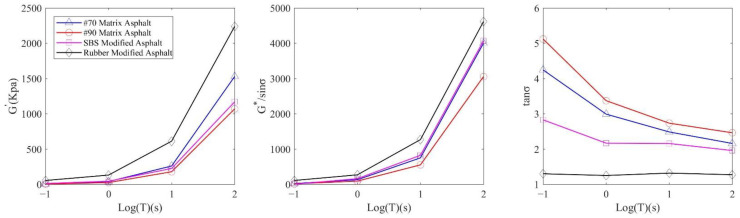
The variation of rheological parameters at the same temperature but different frequencies.

**Figure 7 materials-14-00147-f007:**
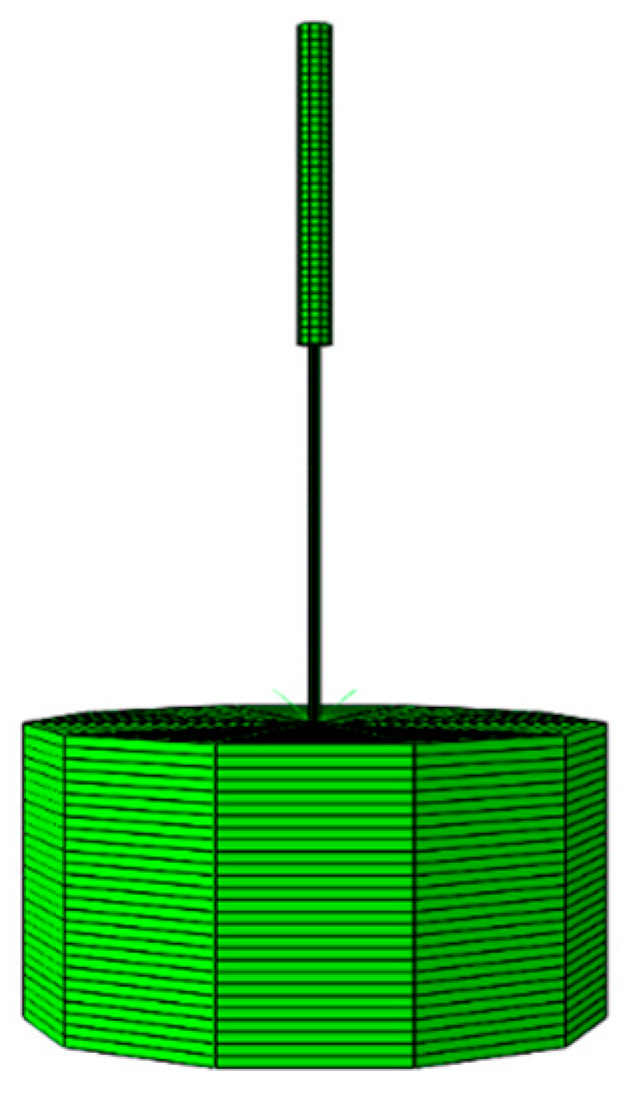
The asphalt penetration model.

**Figure 8 materials-14-00147-f008:**
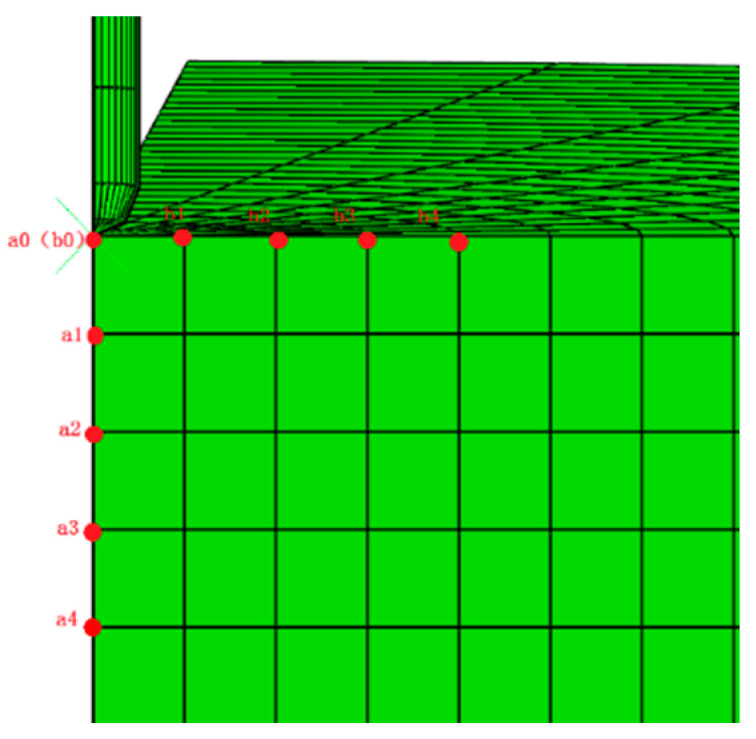
The location of monitoring points.

**Figure 9 materials-14-00147-f009:**
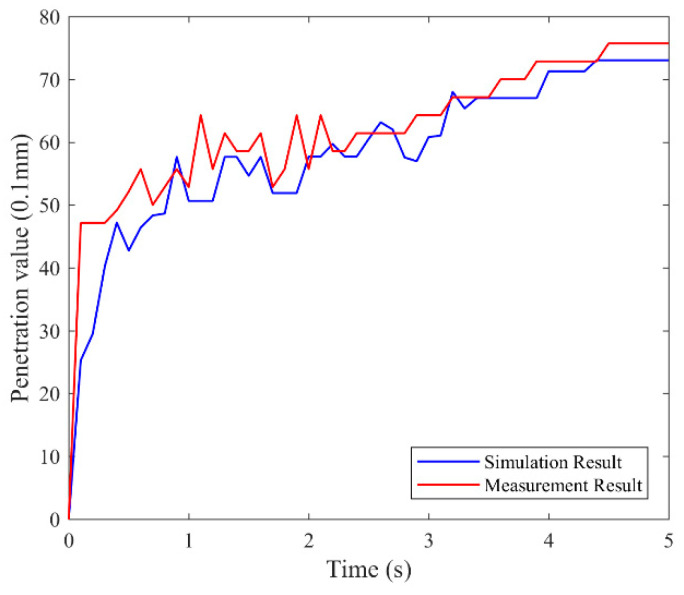
Comparison of simulation result and measurement result.

**Figure 10 materials-14-00147-f010:**
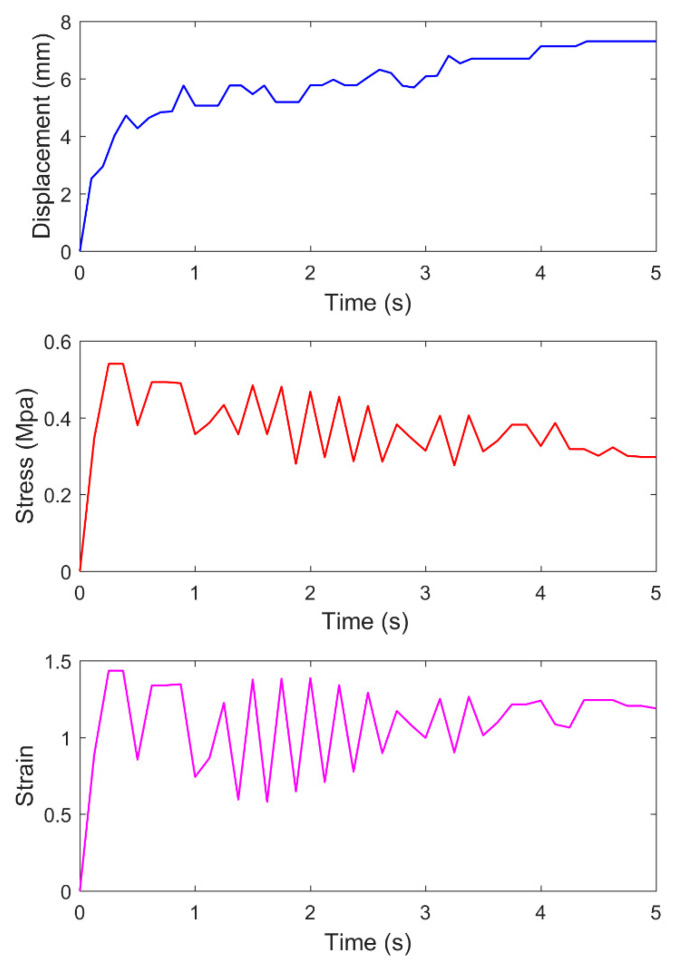
Curves of displacement, stress, and strain of surface monitoring points with time.

**Figure 11 materials-14-00147-f011:**
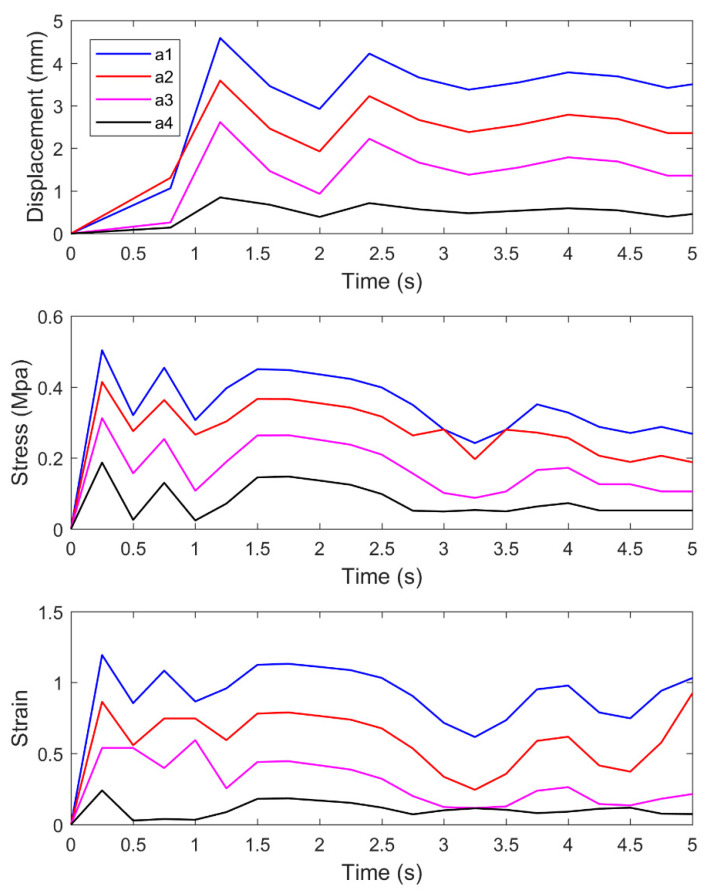
Curves of displacement, stress, and strain of vertical monitoring points with time.

**Figure 12 materials-14-00147-f012:**
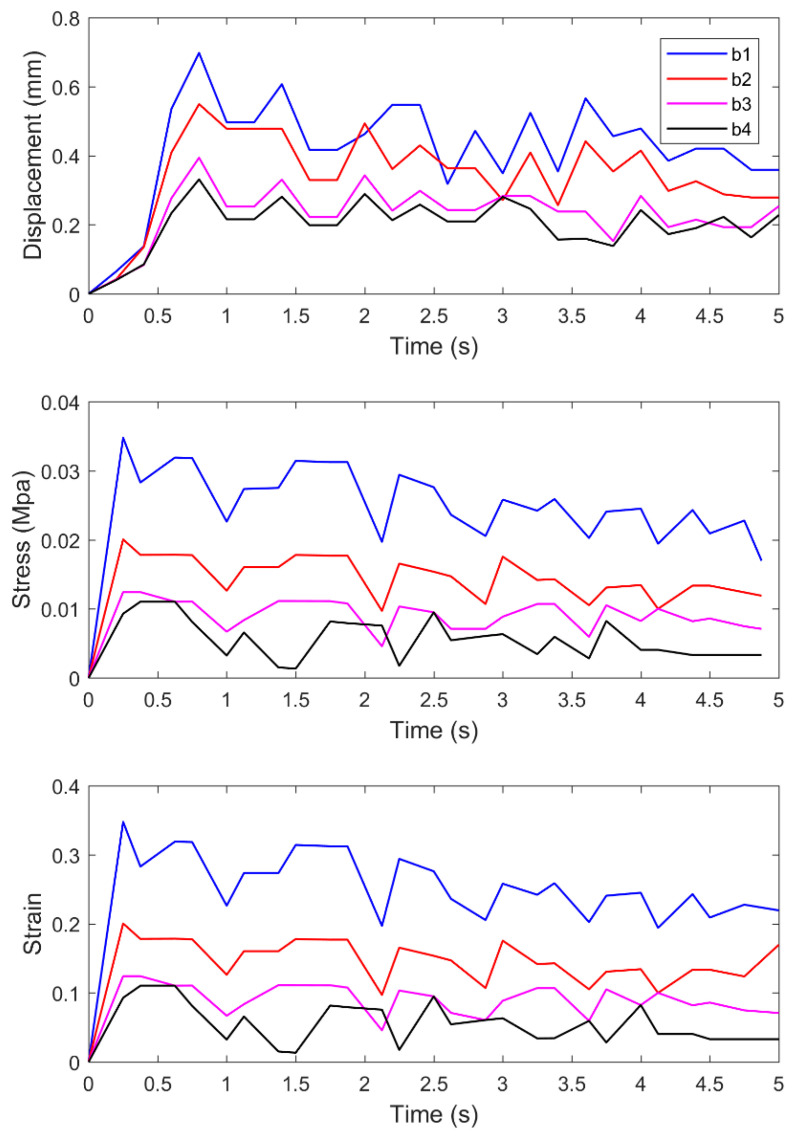
Curves of displacement, stress, and strain of transverse monitoring points with time.

**Table 1 materials-14-00147-t001:** Test parameter setting.

Asphalt Sample	Asphalt Penetration Test Parameter Setting			High-Speed Camera Test Parameter Setting	
Size (mm)	Time (s)	Load (g)	Temperature (°C)	Shooting Time (s)	Shooting Frequency (fps)
Φ55 × 35	5	100	25	5	300

**Table 2 materials-14-00147-t002:** Material parameter setting of the standard needle.

Model	Young’s Modulus Pa	Poisson’s Ratio	Density kg/m^3^
Standard needle	2.068 × 10^11^	0.3	7800

**Table 3 materials-14-00147-t003:** Material parameter setting of the asphalt sample model.

Model	Young’s Modulus Pa	Poisson’s Ratio	Density kg/m^3^	Size
70# matrix asphalt sample	1.7 × 10^6^	0.35	1030	φ 55 mm depth 35 mm

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
