# Peer review of "An Improved Asphalt Penetration Test Method"

_materials, 2020, doi:10.3390/ma14010147_

Round 1

Reviewer 1 Report

The paper is clearly written and reported results. However theoretical interpretation of experimental data does not correspond to the high-quality criteria of the journal of Materials. Therefore I recommend rejecting this manuscript.

Author Response

Thanks for the comment, this paper proposed an improved asphalt penetration test method. The whole process of the asphalt penetration test is analyzed by combining high-speed photography, image processing and numerical simulation methods, which provides abundant test data and a new analytical perspective for the evaluation of asphalt performance.

Reviewer 2 Report

The manuscript “An Improved Asphalt Penetration Test Method” proposes the use of digital processing and image analysis approaches from high speed photographs and the finite element simulation method, to analyze the entire asphalt penetration test process.

The work developed is clearly important from an academic and industrial point of view. In addition, the ideas are very good and the methodology is somewhat innovative.Nevertheless, in my opinion, the manuscript must be rewritten. In particular, the summary is extremely long, confusing and boring.
Authors must also rewrite the conclusions. Right now, this topic presents more evidence than conclusions.

Author Response

Thanks for the reviewer’s comment. According to your comments, we rewrite the abstract and conclusion, delete redundant content, and add DSR experiment to verify the results of the penetration test.

Reviewer 3 Report

The article promises to be interesting.

In the summary, it is better to indicate the practical elements of the new method and future suggestions for improving the method.

The authors have described the shortcomings of the method well and indicate ways to improve.

However, there are a few shortcomings of the article:

- “Through the MATLAB program, the color picture is converted to the grayscale, thus improving the resolution of the image.”

Image conversion does not increase the image resolution - the number of pixels.

The photos shown are not very convincing. Please show zoom - zoom in on the image.

There is a lack of comparing the results with the classical standard measurement method.

The scattering of results for a few samples is not shown.

The discussion and analysis of Figure 5 is valuable and well elaborated.

The model in the article is redundant. It has too little division into elements. There is no analysis of the accuracy of this division into the calculation results. Many other essential elements are missing: gravity, contact analysis, friction, etc. The method for identifying the rheological parameters and calibrating the model is not provided. The results of the calculations in Fig. 10 are questionable.

Such a simplified model should not be shown in the article. Please focus on the method and the analysis of the results for multiple samples. This better relates to the title of the article.

The advantages of the new modified method and guidelines for further research were not indicated. The simplified model does not accurately explain the phenomena occurring in the asphalt during loading.

The conclusions are too general.

Author Response

Thanks for the reviewer’s comments. The replay is attached.

Reviewer 4 Report

The authors proposed an interesting concept of checking the phenomena occurring during the basic asphalt testing. They described it in great detail. They presented conclusions supported by the results achieved.

Author Response

Thanks for your comment!

Round 2

Reviewer 1 Report

* Abstract should be rewritten. The general information should be concisely. Instead, more details of reviewed aspects should be presented.

* Conclusion is shallow, and authors are expected to describe more details.

* Authors could cite the following works in the introduction which is closely related to their work and recently reported:

"Resistance to Ultraviolet Aging of Nano-SiO2 and Rubber Powder Compound Modified Asphalt." Materials 13, no. 22 (2020): 5067.

Warm mix asphalt technology: An up to date review. Journal of Cleaner Production, (2020). 122128.

Ultraviolet aging study on bitumen modified by a composite of clay and fumed silica nanoparticles. Scientific Reports, (2020). 10(1), 1-17.

* Technical terms are misused through the manuscript and the writing needs a revision.

* Validation of models must be present.

Author Response

1. Comment: Abstract should be rewritten. The general information should beconcisely. Instead, more details of reviewed aspects should be presented.

Response: Thanks for the reviewer’s comment. The revised context is as follows:

“Traditional penetration test only measures the total penetration within 5 seconds. The penetration process is not monitored and therefore a large amount of information on the deformation properties of asphalt is not used. This paper documents a study to use high-speed camera to quantify the entire penetration process and use Finite Element Method (FEM) to interpret the penetration process using a viscoelastic model. The penetration-time relationships of several asphalt binders (70#, 90#, a Rubber Modified and a SBS modified binder) have been acquired using the new method and the FEM modeling of the penetration processes is performed. The results show that both stress relaxation and creep appear during the penetration process. The results indicate that the improved test method and its data interpretation procedure may better characterize the properties of asphalt binder, which may extend the applications of the traditional penetration test.”

2. Comment: Conclusion is shallow, and authors are expected to describe more details.

Response: Thanks for the reviewer’s comment. The revised context is as follows:

“1.The improved penetration test using high-speed imaging and FEM modeling has enabled better data interpretation of the traditional penetration test. It may revive the use of penetration test as regular test for asphalt binder characterization.

2. The weaker vibrations during the penetration into rubber modified asphalt and SBS modified asphalt may indicate a strong material structure of the two binders in comparison with the two base asphalt binder 70# and 90#.

3. The viscosity and elasticity of the four kinds of asphalts are quantified in the process of dynamic shear rheological test, which was consistent with the analysis results of the penetration test. In other words, the modified penetration test may have the capability to characterize viscosity and modulus of asphalt binder.

4. It is anticipated that subsequent research to expand the test temperature range, penetration time and applied load, and to improve the FEM model of asphalt penetration may allow to test to become a comprehensive test for asphalt binder.”

3. Comment: Authors could cite the following works in the introduction which is closely related to their work and recently reported.

"Resistance to Ultraviolet Aging of Nano-SiO2 and Rubber Powder Compound Modified Asphalt." Materials 13, no. 22 (2020): 5067.

Warm mix asphalt technology: An up to date review. Journal of Cleaner Production, (2020). 122128.

Ultraviolet aging study on bitumen modified by a composite of clay and fumed silica nanoparticles. Scientific Reports, (2020). 10(1), 1-17”

Response: Thanks for the reviewer’s comment. We have cited these works.

4. Comment: Technical terms are misused through the manuscript and the writing needs a revision.

Response: Thanks for the reviewer’s comment. We checked the manuscript carefully. Redundant content is deleted and revised content is marked in red.

5. Comment: Validation of models must be present.

Response: Thanks for the reviewer’s comment. In order to verify the model, the simulation results of 70# matrix asphalt are compared with the measured results. Figure 10 shows the displacement obtained by the two tests, which is basically in line with the time trends. The simulation method can reflect the change of the asphalt penetration value.

Reviewer 2 Report

  The manuscript has been greatly improved. I accept this way

Author Response

1. Comment: The manuscript has been greatly improved. I accept this way.

Response: Thanks for your comment!

Reviewer 3 Report

The authors improved many elements. The article can be accepted for publication. However, the FEM model must be used with caution.

Author Response

1. Comment: The authors improved many elements. The article can be accepted for publication. However, the FEM model must be used with caution.

Response: Thanks for the reviewer’s comment. We have deleted the redundant content of the FEM part, and point out that the FEM model of asphalt penetration test should be improved in the conclusion.

Round 3

Reviewer 1 Report

I have a problem with the validation of the FEM model, however, the authors tried to solve many problems of the previous copy.

Author Response

Thanks for the reviewer’s comment. In order to validate the FEM model, the simulation results of 70# matrix asphalt are compared with the measured results. The result shows that the displacement obtained by the two tests has the same trends with the time, the simulation method can reflect the change of the asphalt penetration value. In Section 3.3, we explained the reasons for the difference. In the conclusion, we also point out that the FEM model of asphalt penetration test should be improved further.